# Structural Modification and Optimisation of Hyperoside Oriented to Inhibit TGF-β-Induced EMT Activity in Alveolar Epithelial Cells

**DOI:** 10.3390/ph17050584

**Published:** 2024-05-02

**Authors:** Ziye Gao, Mengzhen Xu, Chuanguo Liu, Kai Gong, Xin Yu, Kaihui Lu, Jiang Zhu, Haixing Guan, Qingjun Zhu

**Affiliations:** 1Innovative Institute of Chinese Medicine and Pharmacy, Shandong University of Traditional Chinese Medicine, Jinan 250355, China; 2021111377@sdutcm.edu.cn (Z.G.); 2023100144@sdutcm.edu.cn (M.X.); 2022111508@sdutcm.edu.cn (K.G.); 2022110141@sdutcm.edu.cn (X.Y.); 2022111509@sdutcm.edu.cn (K.L.); 2023111502@sdutcm.edu.cn (J.Z.); 2Experimental Center, Shandong University of Traditional Chinese Medicine, Jinan 250355, China; 60011973@sdutcm.edu.cn; 3Experimental Center, Shandong Provincial Key Laboratory of Traditional Chinese Medicine, Key Laboratory of Traditional Chinese Medicine Classical Theory, Ministry of Education, Shandong University of Traditional Chinese Medicine, Jinan 250355, China; 4Key Laboratory of Traditional Chinese Medicine Classical Theory, Ministry of Education, Shandong University of Traditional Chinese Medicine, Jinan 250355, China

**Keywords:** hypericin derivatives, pulmonary fibrosis, TGF-β1, epithelial–mesenchymal transition (EMT), cell morphology, adhesion and migration

## Abstract

Pulmonary fibrosis (PF) is a disease characterised by diffuse nonspecific alveolar inflammation with interstitial fibrosis, which clinically manifests as dyspnoea and a significant decline in lung function. Many studies have shown that the epithelial–mesenchymal transition (EMT) plays a pivotal role in the pathogenesis of pulmonary fibrosis. Based on our previous findings, hypericin (Hyp) can effectively inhibit the process of the EMT to attenuate lung fibrosis. Therefore, a series of hyperoside derivatives were synthesised via modifying the structure of hyperoside, and subsequently evaluated for A549 cytotoxicity. Among these, the pre-screening of eight derivatives inhibits the EMT. In this study, we evaluated the efficacy of Z6, the most promising hyperoside derivative, in reversing TGF-β1-induced EMTs and inhibiting the EMT-associated migration of A549 cells. After the treatment of A549 cells with Z6 for 48 h, RT-qPCR and Western blot results showed that Z6 inhibited TGF-β1-induced EMTs in epithelial cells by supressing morphological changes in A549 cells, up-regulating E-cadherin (*p* < 0.01, *p* < 0.001), and down-regulating Vimentin (*p* < 0.01, *p* < 0.001). This treatment significantly reduced the mobility of transforming growth factor β1 (TGF-β1)-stimulated cells (*p* < 0.001) as assessed by wound closure, while increasing the adhesion rate of A549 cells (*p* < 0.001). In conclusion, our results suggest that hyperoside derivatives, especially compound Z6, are promising as potential lead compounds for treating pulmonary fibrosis, and therefore deserve further investigation.

## 1. Introduction

Pulmonary fibrosis (PF) is a diffuse interstitial lung disease characterised by inflammation in the lung interstitium, involving alveolar epithelial cells and pulmonary vasculature. It manifests clinically with symptoms like chronic cough and hypoxic respiratory failure. The causative factors and pathogenesis of PF have not yet been fully clarified. Smoking, environmental pollution, occupational exposure to pollutants, and viral infections have all been proposed as risk factors for PF. Emerging evidence suggests that progressive fibrosis is always associated with the epithelial–mesenchymal transition (EMT) of alveolar epithelial cells (AECs) [1,2,3]. Fibrosis is a prevalent pathological mechanism in numerous diseases. With in-depth studies on the mechanisms of fibrosis, we have found that EMTs play a key role in the formation of pulmonary fibrosis [4]. The mechanisms of fibrosis may be a new target for preventing and treating pulmonary fibrosis. The epithelial–mesenchymal transition (EMT) is a disease-developing process that contributes to the pathogenesis of many diseases, including lung cancer and other lung injuries [5,6]. EMT-driven plasticity changes, through transformation into a mesenchymal phenotype, confer metastatic properties to lung epithelial cells or epithelial-derived tumour cells. This change often leads to more severe disease progression and a higher risk of mortality [7]. EMTs can be categorised into three subtypes, depending on the biological setting in which it occurs. One of them, a type 2 EMT, is caused by persistent injury and inflammation, and is an important process of organ fibrosis [8]. Transforming growth factor TGF-β is a key molecule in activating the fibrotic program. TGF-β is up-regulated and activated in fibrotic diseases [9,10,11]. It regulates fibroblast phenotype and function by inducing myofibroblast transdifferentiation, inducing EMTs in alveolar epithelial cells and promoting matrix preservation [12,13,14]. In this process, transforming growth factor β1 (TGF-β1) acts as a pro-fibrotic agent and induces the epithelial–mesenchymal transition (EMT) in alveolar epithelial cells, which results in an elongated spindle-shaped skeleton, reduced adhesion, and the increased migratory capacity of the affected alveolar epithelial cells [15,16].

According to data, severe acute respiratory syndrome coronavirus infection is followed by the severe sequelae of pulmonary fibrosis [17]. Currently, only Pirfenidone and Zidane are approved for the treatment of pulmonary fibrosis [18,19]. However, both have clinical adverse effects, and they can only alleviate pulmonary fibrosis, but cannot reverse lung injury [20,21]. Some studies have shown [10,22] that stem cells and others can treat bleomycin-induced pulmonary fibrosis, but their application effects as well as the mechanism of action need further experimental studies and still have many limitations. Therefore, it is necessary to search for natural, safe, and effective therapeutic agents for treating pulmonary fibrosis [23]. Hypericin is an extremely important class of flavonoids with medicinal functions. In the past few years, hypericin has become a popular traditional Chinese medicine monomer in research. As a potential medicinal ingredient, hypericin has been reported to have a wide range of pharmacological effects against lung cancer, cardiac injury, renal injury, fibrosis, and diabetes mellitus [23,24,25]. It also regulates the immune system, digestive system, and nervous system. Hypericin has a lung-protective function via mitigating bleomycin-induced lung fibrosis through the AKT/GSK3β pathway and reducing levels of MDA, TNF-α, and IL-6 [26]. Meanwhile, some studies have shown that hypericin can exert its anti-hepatocellular carcinoma effect by inhibiting the proliferation of hepatocellular carcinoma cells, breaking their cell cycle, and effectively inhibiting the invasion and metastasis of hepatocellular carcinoma cells [27]. Moreover, through screening and experimental verification, our group has selected several Chinese medicine monomers with antifibrotic properties, among which hypericin has shown better antifibrotic effects through inhibiting EMTs [28].

A549 cells are of type II alveolar epithelial cell origin. They are commonly used for in vitro models to study the process of lung fibrosis after inducing epithelial–mesenchymal transitions using TGF-β1 [29,30,31]. Therefore, we have modified the structure of hyperoside as a lead compound and synthesised a series of derivatives. We evaluated their cytotoxicity effects in A549 cells in vitro. In addition, we identified the potential of hyperoside derivative (Z6) in reversing TGF-β1-stimulated epithelial–mesenchymal transitions (EMTs) and inhibiting EMT-associated adhesion and migration ability in A549 cells. This result has important implications for the development of drug therapy for pulmonary fibrosis. We explored whether synthetic hyperoside derivatives exert antifibrotic effects by inhibiting the TGF-β1-induced EMT process. The aim is to discover novel compounds and derivatives with anti-EMT properties to provide theoretical support for the treatment of EMTs with targeted drugs.

## 2. Results

### 2.1. Chemistry

For the basic structure of hyperoside, this paper focuses on the structural modification of two parts. Molecular docking revealed that the hydroxyl group of the A ring No. 5 points to the outside of the active pocket [32], which is not directly related to its main activity. Therefore, it can be eliminated from the structure. Our group have previously studied the anti-EMT activity of hypericin with saccharin and hesperidin, and found that quercetin with the B ring attached to the second position was not effective against EMTs, so we designed the synthesis to be modified at the third position. Based on the advantages of hyperoside itself acting as a group and its poor water solubility affecting the efficacy of the drug, we first maintained the basic structural skeleton of hyperoside, keeping the C ring unchanged. Then, we divided it into two series of compounds according to the presence or absence of the hydroxyl group at position 7 of the A ring, the type of the substituent of the B ring, and the position of the linkage [33,34]. In addition to considering its poor water solubility and lipid solubility, as well as the specificity of the hydroxyl group [35], we also introduced halogenated elements and methoxy groups at different sites [36].

In the present study, deoxybenzoic acid was obtained by reacting resorcinol or phenol with phenylacetic acid containing the corresponding substituent groups under nitrogen in the presence of BF_3_·Et_2_O, followed by treatment with Weissmayer’s reagent formed with methane sulfonyl chloride and DMF (N, N-dimethylformamide) in the presence of BF_3_·Et_2_O to obtain the final products, respectively, which were then obtained using the mild reaction conditions, short reaction time, and excellent yields. The structures of the target compounds were confirmed with ^1^H NMR and ^13^C NMR.

Compound G0: yellow solid, yield 64%. ^1^H NMR (600 MHz, DMSO-d6) δ 10.76 (s, 1H), 8.99 (s, 1H), 8.95 (s, 1H), 8.25 (s, 1H), 7.96 (d, J = 8.7 Hz, 1H), 7.01 (d, J = 2.1 Hz, 1H), 6.93 (dd, J = 8.7, 2.3 Hz, 1H), 6.85 (d, J = 2.2 Hz, 1H), 6.80 (dd, J = 8.1, 2.1 Hz, 1H), 6.76 (d, J = 8.2 Hz, 1H). ^13^C NMR (151 MHz, DMSO-d6) δ 174.04, 161.84, 156.73, 152.14, 144.63, 144.15, 126.67, 123.00, 122.37, 119.21, 116.03, 115.97, 114.65, 114.46, 101.44.

Compound G1: white solid, yield 83%. ^1^H NMR (600 MHz, DMSO-d6) δ 10.82 (s, 1H), 10.30 (s, 1H), 8.38 (s, 1H), 7.96 (d, J = 4.0 Hz, 1H), 7.59 (s, 1H), 7.36 (d, J = 10.5 Hz, 1H), 7.01 (d, J = 8.4 Hz, 1H), 6.95 (dd, J = 8.8, 2.2 Hz, 1H), 6.88 (d, J = 2.2 Hz, 1H). ^13^C NMR (151 MHz, DMSO-d6) δ 172.64, 160.84, 160.51, 155.62, 151.65, 150.96, 128.30, 126.73, 125.50, 122.08, 120.36, 117.48, 114.73, 113.46, 100.35.

Compound G2: yellow solid, yield 58%. ^1^H NMR (600 MHz, DMSO-d6) δ 10.80 (s, 1H), 8.48 (s, 1H), 8.38 (s, 1H), 7.98 (d, J = 8.8 Hz, 1H), 6.95 (d, J = 11.2 Hz, 1H), 6.88 (s, 3H), 3.79 (s, 6H). ^13^C NMR (151 MHz, DMSO-d6) δ 175.08, 162.99, 157.79, 153.78, 148.12, 136.05, 127.79, 124.01, 122.44, 117.12, 115.64, 107.13, 102.56, 56.55, 49.07.

Compound G3: light yellow solid, yield 57%. ^1^H NMR (600 MHz, DMSO-d6) δ 10.82 (s, 1H), 9.98 (s, 1H), 8.38 (s, 1H), 7.97 (d, J = 8.8 Hz, 1H), 7.42 (dd, J = 12.7, 2.0 Hz, 1H), 7.23 (d, J = 8.3 Hz, 1H), 7.01–6.93 (m, 2H), 6.87 (d, J = 2.2 Hz, 1H). ^13^C NMR (151 MHz, DMSO-d6) δ 174.91, 163.10, 157.85, 153.93, 150.19, 145.10, 145.02, 127.78, 125.47, 125.45, 123.77, 123.73, 122.73, 117.83, 117.81, 117.12, 117.01, 116.99, 115.72, 102.60.

Compound G4: white solid, yield 62%. ^1^H NMR (600 MHz, DMSO-d6) δ 10.78 (s, 1H), 9.53 (s, 1H), 8.29 (s, 1H), 7.97 (d, J = 8.7 Hz, 1H), 7.54–7.27 (m, 2H), 6.94 (dd, J = 8.7, 2.3 Hz, 1H), 6.86 (d, J = 2.3 Hz, 1H), 6.84–6.79 (m, 2H). ^13^C NMR (151 MHz, DMSO-d6) δ 175.16, 162.96, 157.89, 157.64, 153.29, 130.54, 127.76, 123.95, 123.01, 117.10, 115.59, 115.41, 102.56.

Compound G5: light red solid, yield 49%. ^1^H NMR (600 MHz, DMSO-d6) δ 10.81 (s, 1H), 9.45 (s, 1H), 8.35 (s, 1H), 7.97 (t, J = 9.5 Hz, 1H), 7.21 (t, J = 7.9 Hz, 1H), 7.03–7.00 (m, 1H), 6.95 (dd, J = 8.7, 2.2 Hz, 2H), 6.88 (d, J = 2.2 Hz, 1H), 6.79–6.75 (m, 1H). ^13^C NMR (151 MHz, DMSO-d6) δ 173.77, 162.02, 156.79, 156.43, 153.09, 132.67, 128.49, 126.74, 123.00, 118.88, 116.06, 115.45, 114.63, 114.12, 101.55, 48.00, 35.18.

Compound G6:white solid, yield 47%. ^1^H NMR (600 MHz, DMSO-d6) δ 10.79 (s, 1H), 9.02 (s, 1H), 8.29 (s, 1H), 7.97 (d, J = 8.8 Hz, 1H), 7.06 (s, 1H), 6.96 (s, 1H), 6.95 (s, 1H), 6.95–6.92 (m, 1H), 6.87 (d, J = 2.2 Hz, 1H), 3.80 (s, 3H). ^13^C NMR (151 MHz, DMSO-d6) δ 175.05, 162.99, 157.85, 153.54, 147.96, 146.49, 127.78, 125.15, 123.82, 120.16, 117.12, 116.91, 115.62, 112.40, 102.57, 56.13, 40.52.

Compound G7: white solid, yield 45%. ^1^H NMR (600 MHz, DMSO-d6) δ 10.79 (s, 1H), 9.09 (s, 1H), 8.33 (s, 1H), 7.98 (d, J = 8.8 Hz, 1H), 7.17 (d, J = 1.8 Hz, 1H), 6.99 (dd, J = 8.1, 1.9 Hz, 1H), 6.94 (dd, J = 8.8, 2.2 Hz, 1H), 6.87 (d, J = 2.2 Hz, 1H), 6.82 (d, J = 8.1 Hz, 1H), 3.80 (s, 3H). ^13^C NMR (151 MHz, DMSO-d6) δ 175.15, 162.99, 157.86, 153.55, 147.63, 146.94, 127.79, 123.99, 123.47, 122.00, 117.13, 115.66, 115.63, 113.73, 102.57, 56.14, 40.53.

Compound G8: white solid, yield 83%. ^1^H NMR (600 MHz, DMSO-d6) δ 10.80 (s, 1H), 8.34 (s, 1H), 7.98 (d, J = 8.7 Hz, 1H), 7.66–7.38 (m, 2H), 7.01–6.98 (m, 2H), 6.95 (dd, J = 8.8, 2.3 Hz, 1H), 6.87 (d, J = 2.2 Hz, 1H), 3.79 (s, 3H). ^13^C NMR (151 MHz, DMSO-d6) δ 175.08, 163.03, 159.42, 157.92, 153.63, 130.55, 127.77, 124.70, 123.62, 117.09, 115.65, 114.07, 102.60, 55.61.

Compound G9: white solid, yield 88%. ^1^H NMR (600 MHz, DMSO-d6) δ 10.84 (s, 1H), 8.41 (s, 1H), 7.99 (d, J = 8.8 Hz, 1H), 7.34 (t, J = 7.9 Hz, 1H), 7.17–7.12 (m, 2H), 6.95 (D, J = 8.3, 4.3, 1.6 Hz, 2H), 6.89 (d, J = 2.2 Hz, 1H), 3.79 (s, 3H). ^13^C NMR (151 MHz, DMSO-d6) δ 174.83, 163.15, 159.47, 157.89, 154.48, 133.91, 129.62, 127.83, 123.80, 121.66, 117.11, 115.77, 115.10, 113.73, 102.65, 55.56.

Compound G10: white solid, yield 41%. ^1^H NMR (600 MHz, DMSO-d6) δ 10.87 (s, 1H), 8.49 (s, 1H), 7.98 (d, J = 8.8 Hz, 1H), 7.81 (t, J = 1.8 Hz, 1H), 7.70–7.44 (m, 2H), 7.40 (t, J = 7.9 Hz, 1H), 6.97 (dd, J = 8.7, 2.3 Hz, 1H), 6.90 (d, J = 2.2 Hz, 1H). ^13^C NMR (151 MHz, DMSO-d6) δ 174.58, 163.29, 157.91, 155.02, 135.01, 131.88, 130.96, 130.72, 128.36, 127.81, 122.52, 121.79, 116.98, 115.89, 102.71.

Compound Z1: solid, yield 61%. ^1^H NMR (600 MHz, DMSO-d6) δ 10.35 (s, 1H), 8.55 (s, 1H), 8.15 (dd, J = 7.9, 1.4 Hz, 1H), 7.86–7.81 (m, 1H), 7.69 (d, J = 8.4 Hz, 1H), 7.63 (d, J = 2.1 Hz, 1H), 7.53 (t, J = 7.5 Hz, 1H), 7.40 (dd, J = 8.4, 2.1 Hz, 1H), 7.03 (d, J = 8.4 Hz, 1H). ^13^C NMR (151 MHz, DMSO-d6) δ 174.56, 154.98, 153.69, 152.30, 133.60, 129.50, 127.95, 124.95, 124.90, 123.12, 123.00, 121.96, 118.75, 117.79, 115.70.

Compound Z3: white solid, yield 47%. ^1^H NMR (600 MHz, DMSO-d6) δ 8.65 (s, 1H), 8.16 (dd, J = 8.0, 1.5 Hz, 1H), 7.88–7.85 (m, 1H), 7.84 (d, J = 1.8 Hz, 1H), 7.72 (d, J = 8.1 Hz, 1H), 7.63 (d, J = 7.7 Hz, 1H), 7.60 (D, J = 8.0, 1.9, 0.9 Hz, 1H), 7.56–7.52 (m, 1H), 7.43 (t, J = 7.9 Hz, 1H). ^13^C NMR (151 MHz, DMSO-d6) δ 175.37, 156.10, 155.87, 134.91, 134.77, 131.90, 131.16, 130.82, 128.41, 126.24, 126.01, 124.24, 122.92, 121.88, 118.95.

Compound Z5: white solid, yield 34%. ^1^H NMR (600 MHz, DMSO-d6) δ 10.82 (s, 1H), 8.41 (s, 1H), 7.98 (d, J = 8.7 Hz, 1H), 7.32 (t, J = 7.9 Hz, 1H), 7.15–7.11 (m, 2H), 6.96–6.92 (m, 2H), 6.89 (d, J = 2.3 Hz, 1H), 4.06 (d, J = 6.9 Hz, 1H). ^13^C NMR (151 MHz, DMSO-d6) δ 173.76, 162.01, 161.69, 156.78, 156.41, 153.08, 132.66, 128.47, 126.72, 122.98, 118.87, 116.05, 115.44, 114.62, 114.10, 101.54.

Compound Z6: white solid, yield 60%. ^1^H NMR (600 MHz, DMSO-d6) δ 9.05 (s, 1H), 8.47 (s, 1H), 8.16–8.13 (m, 1H), 7.84–7.80 (m, 1H), 7.68 (d, J = 8.4 Hz, 1H), 7.52 (t, J = 7.5 Hz, 1H), 7.09 (s, 1H), 6.99 (s, 1H), 6.97 (s, 1H), 3.80 (s, 3H). ^13^C NMR (151 MHz, DMSO-d6) δ 175.85, 156.04, 154.41, 147.71, 147.09, 134.55, 126.00, 125.91, 124.38, 124.29, 123.20, 122.05, 118.84, 115.71, 113.68, 56.17.

Compound Z7: light solid, yield 26%. ^1^H NMR (600 MHz, DMSO-d6) δ 9.13 (s, 1H), 8.51 (s, 1H), 8.15 (dd, J = 8.0, 1.6 Hz, 1H), 7.84–7.80 (m, 1H), 7.69 (d, J = 8.2 Hz, 1H), 7.53–7.50 (m, 1H), 7.20 (d, J = 2.0 Hz, 1H), 7.04 (dd, J = 8.1, 2.0 Hz, 1H), 6.84 (d, J = 8.1 Hz, 1H), 3.81 (s, 3H). ^13^C NMR (151 MHz, DMSO-d6) δ 175.85, 156.04, 154.41, 147.71, 147.09, 134.55, 126.00, 125.91, 124.38, 124.29, 123.20, 122.05, 118.84, 115.71, 113.68, 56.17.

### 2.2. Biological Evaluation

#### 2.2.1. Effects of Hyperoside Derivatives on Cell Viability

The anti-pulmonary fibrosis drugs that we hope to modify in the experiment must first be specific, well tolerated during the experiment, and exert their effects stably and effectively without affecting the growth of normal alveolar epithelial cells. Therefore, before the evaluation of the effect of anti-pulmonary fibrosis drugs, a preliminary screening of the compound concentrations was first carried out using the effect of the drugs on the survival rate of alveolar epithelial cells. The concentrations without significant inhibitory effects on alveolar epithelial cells were screened for subsequent anti-fibrotic effects. In this experimental study, the sensitivity of the cells to the compounds was determined using the MTT assay to determine the effect of the compounds on cell viability. The experimentally synthesised hyperoside derivatives were configured into 10 mmol/L and 40 mmol/L solutions with dimethyl sulfoxide (DMSO), respectively, and stored at −20 °C for backup. A concentration gradient in the range of 0–400 μmol/L was set up to measure the survival rate of A549 cells at each concentration, and the concentration of IC_10_ was obtained (Table 1).

The preliminary screening was conducted according to the experimental results of MTT, and the effect on the survival rate of A549 cells was analysed. In this paper, the subsequent dosage concentration was discussed based on the experimental data of the MTT concentration gradient. The results showed that some compounds did not show obvious inhibition in a certain range of concentrations, but these compounds either showed a certain inhibition in the concentration range of 50–400 μmol/L, or showed a concentration-dependent inhibition in the concentration range. According to the preliminary experimental study of the research group, the drug concentration was 10 μmol/L, and although it displayed an anti-fibrosis effect, the effect was not obvious. In the later stage, hypericin and 16 compounds were determined using MTT, and the concentration of IC_10_ was roughly 30 μmol/L, following which, the concentration of the drug increased to 20 μmol/L and 30 μmol/L, respectively. After observing that some compounds were too lethal to cells at 30 μmol/L, it was finally determined that the dosage concentration was 20 μmol/L. The subsequent experimental study on the anti-pulmonary fibrosis effect of hypericin and 16 derivatives through inhibiting the TGF-β1-induced EMT process was conducted.

#### 2.2.2. Preliminary Screening of Hyperoside and Its Derivatives with Anti-EMT Activity

The cell morphology was observed under the 10× objective of the inverted phase contrast microscope. The results were shown in Figure 1A. The cell morphology of the blank control group was complete, with tight connections between the cells, which became cobblestone-like typical epithelial cell morphology. After the TGF-β1 induction, the cell morphology gradually changed from cobblestone-like polygonal to long spindle-shaped fibroblasts, and the cell-to-cell gap was gradually enlarged. When compared with the model group, the cells in the hypericin group had a shortened cell morphology, tight inter-cellular junctions, and smaller gaps. When compared with the cells in the hypericin group, the cell morphology of G0, G3, G9, G10, Z1, Z3, Z5, and Z6 changed after the intervention, the cells showed cobblestone-like cell shapes, and the inter-cellular gaps became smaller; whereas, after the intervention of the other compounds, the cell morphology was not significantly shortened, the inter-cellular gaps were loose, and there was no apparent cobblestone-like morphology. Preliminary evidence suggests that these candidate derivatives inhibit TGF-β1-induced EMTs.

To investigate whether the 16 derivatives inhibited TGF-β1-induced EMTs in A549 cells, we examined the effects of the compounds on EMT-associated mRNAs in TGF-β1-induced A549 cells using the qRT-PCR assay. EMTs play a vital role in the formation of organ fibrosis [37,38,39,40,41]. It has been shown that the down-regulation of the epithelial marker E-cadherin, an important intercellular adhesion protein, is one of the hallmarks of the EMT process. In addition, the up-regulation of the mesenchymal marker waveform protein is also a hallmark of the EMT process [39,42,43,44]. Therefore, the inhibitory effect of hyperoside and the 16 derivatives of the EMT process can be assessed by detecting the changes in the expression of the above markers. We initially showed that some compounds were able to alter the morphology of TGF-β1-induced A549 cells to different degrees. G0, G9, G10, Z1, Z3, Z5, Z6, and other compounds were able to increase the expression of E-cadherin mRNA and decrease the expression of Vimentin mRNA in TGF-β1-induced A549 cells (Figure 1B,C). During the screening process, we found that the anti-EMT activity of hyperoside was increased after the removal of the hydroxyl group at position 7 of the A ring. The activity was increased after the introduction of halogenated elements as well as methoxy at position 3 of the B ring. It was found that the level of pharmacological activity was highly related to the number and position of the substitution of the methoxy group. Generally, the methoxy group on the B ring would have a close relationship with the increase in activity. This point of view also verified the results of our preliminary screening. After analysing the data and so on, we chose eight derivatives from the two series of compounds and continued the post-experiment.

#### 2.2.3. Validation of Hyperoside and Its Derivatives with Anti-EMT Activity

After the candidate compounds were identified, their ability to combat pulmonary fibrosis was investigated using Western blot experiments. Since an important part of the disease process of pulmonary fibrosis is cell activation, TGF- β1 was used to stimulate the activation of A549 cells to construct a cell model of pulmonary fibrosis. TGF-β1-induced EMT not only induced changes in the morphology of the lung epithelial cells, but also caused changes in the EMT-related markers and cell function. When the EMT occurs, the most significant changes in the expression level of intracellular proteins include the increased expression of E-cadherin (Figure 2A,B) and the decreased expression of Vimentin (Figure 2C,D). These proteins, as important factors regulating the process of the EMT, are also of great significance in terms of their expression levels. When combined with the results of the qRT-PCR quantitative analysis, we found that the eight canaryl glucoside derivatives were able to increase the relative expression of E-cadherin mRNA and protein to different degrees. At the same time, they decreased the relative expression of Vimentin mRNA and protein.

The EMT process is characterised by a phenotypic transformation of epithelial cells to mesenchymal cells, which occurs at high rates in fibrotic tissues and cancers. This transformation increases the invasive capacity and migratory potential of the cells. It is characteristic of a worsening disease process, and it accelerates drug resistance in cancer [45,46,47,48].

We believe that the preliminarily screened myricetin derivatives can alter the expression of EMT-related markers in TGF-β1-induced A549 cells, and can delay pulmonary fibrosis by targeting EMTs. Our modified myricetin significantly altered the expression levels of E-cadherin and Vimentin, proving that the synthesised myricetin derivatives could inhibit the EMT process at the molecular level.

Among these hyperoside derivatives, Z6 was the most promising to act as an anti-pulmonary fibrosis agent without affecting the growth of A549 cells stably and effectively. Therefore, to determine whether compound Z6 acts as an inhibitory compound for EMTs in epithelial cells. We chose Z6 to validate its anti-EMT activity against TGF-β1-induced A549 cells. Based on the previous experiments, Z6 at a concentration of 20 μM was selected to further validate its action as an anti-EMT agent by studying its effect on the adhesion ability in TGF-β1-stimulated A549 cells.

The loss of cell–cell adhesion is one of the causes of excessive epithelial cell migration. Given the promising effect of Z6 in inhibiting TGF-β1-induced EMT in epithelial cells, we investigated whether Z6 affects the adhesion ability of A549 cells. To this end, we performed a cell adhesion assay to assess whether Z6 could increase the adhesion capacity of A549 cells (Figure 3A,B).

Given the promising results of Z6 in inhibiting the TGF-β1-induced EMT in epithelial cells, we investigated whether Z6 affects the EMT-related migratory capacity in A549 cells. To this end, an in vitro wound healing assay was performed to assess whether Z6 acts as an anti-metastatic agent in A549 cells (Figure 4A,B).

As shown in Figure 4A,B, TGF-β1-treated cells exhibited increased wound closure at 48 h when compared with cells not treated with TGF-β1. The treatment of cells with 20 μM Z6 for 48 h significantly reduced the migration of TGF-β1-stimulated cells, a phenomenon confirmed through the qualitative assessment of wound closure.

## 3. Discussion

In this study, we designed and synthesised 16 hyperoside derivatives and assayed their cytotoxicity. Subsequently, EMTs were induced in A549 cells using TGF-β1. The expression of EMT-related marker genes E-cadherin and Vimentin was detected with qRT-PCR. The results showed that hyperoside derivatives G0, G3, G9, G10, Z1, Z3, Z5, and Z6 could significantly increase the expression of E-cadherin and decrease the expression of Vimentin. The results of the Western blot experiments also proved this. From the above results, we concluded that all of the above eight derivatives exhibited anti-EMT activity, and the most effective one was Z6. Therefore, based on the above experiments, we continued the cell function assay of Z6. The results showed that Z6 could significantly enhance cell adhesion and reduce cell migration. Thus, it can be seen that Z6 has good anti-EMT activity and can delay the TGF-β1-induced EMT process.

Pulmonary fibrosis is a disease that can cause serious damage to people’s health, and is no less harmful than oncological diseases [4,6]. Since the prevalence of C pneumonia, patients have shown symptoms of pulmonary fibrosis to varying degrees, making the treatment of pulmonary fibrosis a focus of research. According to numerous studies, the process of a epithelial–mesenchymal transition (EMT) is one of the most important aspects in the pathogenesis of pulmonary fibrosis. EMTs confer metastatic properties on lung epithelial cells. This often indicates more severe disease progression and a higher risk of mortality [42,49]. Many growth factors are currently known to induce EMTs, among which the transforming growth factor TGF-β is considered a key molecule in the initiation of the fibrotic program [9]. TGF-β is up-regulated and activated in fibrotic diseases, and it is capable of inducing pulmonary fibrosis through the stimulation of both classical and non-classical signalling pathways, which triggers the activation of myofibroblasts and the overexpression of the extracellular matrix. TGF-β is a key molecule for the initiation of the fibrotic program [10,16].

Natural compounds are abundant and diverse, and they are an extremely important class of compounds with useful functions, having many applications in anti-inflammatory, antifibrotic, and anticancer fields [23,24]. We have synthesised two series of compounds through modifying hyperoside, linking bioactive groups to the A and B rings of the parent nucleus, introducing electron-withdrawing groups such as halogen and carboxylic acid at different sites, and introducing electron-donating groups such as methoxy to improve the lipid- and water-solubility of the drug [33,36]. Our preliminary findings suggest that synthesised hypericin derivatives may alleviate pulmonary fibrosis via inhibiting the EMT process. The expression of signature proteins involved in the EMT directly influences the degradation of transition deposition, facilitating the reversal of pulmonary fibrosis by inhibiting the activation of A549 cells. Thus, these derivatives hold promise for playing an anti-pulmonary fibrosis role. It was found that the presence or absence of the hydroxyl group on the A ring also played an essential role in the inhibition of the EMT process, and the substituent group on the B ring displayed the activity of methoxy > halogenated elements for the inhibition of EMTs. In conclusion, the active hyperoside derivative Z6 was obtained with structural modification, which provides a lead compound for the research and development of derivatives as targeted drugs for treating pulmonary fibrosis. The search for novel and effective compounds that can be used to treat pulmonary fibrosis remains a major scientific challenge. Further studies are needed to determine the feasibility of Z6 as a treatment for pulmonary fibrosis.

## 4. Materials and Methods

### 4.1. Chemical Methods

All commercially available solvents, substrates, and reagents were used without further purification. Thin layer chromatography (TLC) was performed on TCL Silica gel 60 F254 (0.20 mm; Qingdao Ocean Chemical Factory, Qingdao, China). Thin layer chromatography (TLC) was performed on TCL Silica gel 60 F254 (0.20 mm; Qingdao Ocean Chemical Factory, China). Chemical HG/T2354-92 silica gel (200–300 mesh, Qingdao) was used for chromatography. Spots on TLC plates were visualised with UV light. ^1^H and ^13^C nuclear magnetic resonance (NMR) spectra were recorded on a Bruker Avance 400 spectrometer (Bruker Company, Bremen, Germany) with DMS0-d6 as the solvent.

### 4.2. Synthesis

The phenol derivative (1eq, 5 mmol) and acetic acid derivative (1eq, 5 mmol) were weighed and added to a 500 mL round bottom flask, to which we added boron trifluoride ethyl ether solution (2 mL) at a concentration of 46.5%, and then was heated under nitrogen at 110 °C for 4 h for the reaction. The heating was completed and cooled to room temperature. Methyl sulfonyl chloride (MeSO_2_Cl) solution (4.5 mL) and DMF (22 mL) were added to the magnet in advance at room temperature and stirred for 30 min; vilsmeier reagent, stirring, and mixing were completed within 15 min, added drop by drop to the round-bottomed flask. When the dropwise addition was completed, the temperature was raised to 110 ℃, and the reaction continued under nitrogen conditions for 4–7 h. The reaction was detected using the TLC method. After the reaction was completed, 20 mL of pre-cooled distilled water was added to the round-bottomed flask, which was then poured into a separatory funnel; 80 mL of ethyl acetate was then added, and the extraction was carried out to collect the upper layer of liquid in the separatory funnel. The combined organic layer was washed with H_2_O (100 mL), dried over anhydrous Na_2_SO_4_, and evaporated until dry. TLC was performed to detect the reaction, and after the reaction was fully detected via thin layer chromatography, it was purified by silica gel column (DCM:CH_3_OH = 75:1→10:1) to obtain the corresponding hyperoside derivatives (Figure 1A,B).

### 4.3. Cell Culture and Drug Configuration

A549 cells (Procell, Wuhan, China) were cultured in an incubator at 37℃ in 5% CO_2_ with RPMI-1640 (Gibco, Sigma Aldrich, Søborg, Denmark) containing 10% foetal bovine serum (Gibco, Sigma Aldrich, Denmark) and 1% penicillin-streptomycin (Gibco, Sigma Aldrich, Denmark). The experimentally synthesised G-series and Z-series compounds were configured into 10 mmol/L and 40 mmol/L solutions with dimethyl sulfoxide (DMSO), respectively, and stored at −20 °C.

#### 4.3.1. Effect of Compounds on A549 Cell Activity by MTT Assay

The effect of hyperoside derivatives on the activity of A549 cells was determined with the MTT assay. The cell density was adjusted to 5 × 10^4^ cells/mL, and 150 uL per well was added to a 96-well plate (PerkinElmer, Waltham, MA, USA) and placed in the incubator for 24 h. Derivatives were added to the 96-well plate to make the concentration 0, 25, 50, 100, 200, 400 umol/L, and continued to incubate in the incubator for 48 h. Furthermore, 90 uL of the culture solution and 10 uL of the MTT solution (Solabio, Beijing, China) were added to each well, incubated for 4 h under light protection, and then the incubation was terminated. Additionally, 110 µL of the Formazan dissolution solution was added to each well after aspirating the supernatant. The solution was dissolved by shaking on a shaking table at a low speed for 10 min, and the absorbance (OD) of each well was measured at 490 nm. The OD of each well was measured at 490 nm, and the cell viability was calculated according to the formula.

#### 4.3.2. qPCR Assay

A549 cells were inoculated in 6-well plates at a cell density of 1.0 × 10^5^ cells/mL and were then incubated in a cell culture incubator for 24 h for subsequent priming. Cells were divided into three groups: control (cell culture medium containing 1% FBS and no TGF-β1); TGF-β treated (cell culture medium containing 1% FBS and 5 ng/mL TGF-β1); and compound treatment (cell culture medium containing 1% FBS with 5 ng/mL TGF-β1 and 20 μM monomeric compounds), and were then incubated for 48 h. Total RNA was extracted from A549 cells using an RNA extraction kit (Vazyme, Nanjing, China) according to the manufacturer’s instructions, and the RNA was reverse transcribed into cDNA using a HiScript III RT SuperMix for qPCR kit with gDNA wipe (Vazyme, Nanjing, China). The mRNA expression levels were determined with qRT-PCR using a QuantStudio™ 5 qRT-PCR system (Thermo Fisher, Waltham, MA, USA). E-cadherin waveform protein expression levels were normalised to β-actin. Primer sequences are listed in Table 2.

#### 4.3.3. Western Blot

When A549 cells grew to about 80–90%, they were washed three times with PBS (Solarbio, Beijing, China) and digested with 0.25% trypsin (Gibco, Sigma Aldrich, Denmark). Then, they were inoculated in six-well plates at a density of 150,000 cells/well, and then incubated in a cell culture incubator for 24 h for subsequent priming. Set the groups as follows: control (cell culture medium containing 1% FBS and no TGF-β1); TGF-β treated (cell culture medium containing 1% FBS and 5 ng/mL TGF-β1); and compound treatment (cell culture medium containing 1% FBS with 5 ng/mL TGF-β1 and 20 μM monomeric compounds). Cells were treated for 48 h. Protein samples were collected using lysing cells with RIPA containing 1% protease inhibitor (PMSF) and 1% phosphatase inhibitor (Beyotime, Beijing, China) for 30 min on ice. The cell suspension was shaken every 5 min. The cell suspension was centrifuged at 13,000 rpm for 25 min at 4℃. After centrifugation, the protein-containing supernatant was slowly aspirated and denatured in a dry metal bath at 100 °C for 10 min, and then stored at −20 °C for later use. Furthermore, 35 μg of total protein was separated using 10% separating gel and 5% concentrating gel. The proteins were transferred onto 0.45 μm PVDF membranes (Millipore, MA, USA) and closed via slow shaking for 2 h in 10% skim milk powder. The PVDF membrane was then washed five times in a TBST buffer solution for 5 min each time, and the membrane was blocked with the primary antibody at a 1:5000 ratio overnight at 4 °C. The PVDF membrane was washed with the TBST buffer solution and incubated with a secondary antibody at 1:20,000 on a shaker for approximately 1.5 h at room temperature. The membrane was analysed using an automated chemiluminescence imaging analyser (Tanon, Shanghai, China) to analyze the immunoblots, and the grey values were analysed using densitometry (Image J 1.8.0 software).

#### 4.3.4. Determination of Adhesion

A549 cells (2 × 10^5^ cells/mL) were inoculated into 6-well plates and incubated for 24 h. Then, three subgroups were set up as follows: the control group (cell culture medium with 1% FBS and no TGF-β1); the TGF-β treatment group (cell culture medium with 1% FBS and 5 ng/mL TGF-β1); and the compound treatment group (cell culture medium with 1% FBS and 5 ng/mL TGF-β1 and 20 μM Z6 compound), and the cells were further incubated for 48 h. The cells were then incubated for 2 h in the RPIM1640 medium. Fibronectin (FN; 0.2 mg/mL, Solarbio, Beijing, China) was diluted with PBS into 100 uL per well and punched into 96-well plates to a final concentration of 2 μg/cm^2^, dried in an oven for 60 min, and left at 4 °C overnight. Cells were harvested from 6-well plates, collected and inoculated into 96-well plates, and incubated in an incubator for 45 min for staining. Cells were fixed with a 4% formaldehyde solution for 10 min at room temperature, and permeabilised with 0.5% Triton X-100 solution for 5 min. Cell nuclei were stained with 50 μL of Hoechst 33342 at a concentration of 10 μg/mL 30. Images were captured using a fluorescence microscope with an objective lens of 20×. Pictures of the number of cells adhering after treatment with different monomer compounds were analysed using Image J.

#### 4.3.5. Cell Migration

A549 cells (1.5 × 10^5^ cells/mL) were inoculated into 6-well plates (Corning, NY, USA) and cultured with a serum-free cell culture medium for 24 h. Furthermore, 200 μL of sterilised lance tip was taken and scratched vertically according to the previous pen marks to construct an in vitro scratch model, and the same strength and width were controlled in each well; after the scratching was completed, it was washed with PBS buffer until the scratched area had no cell residue, and the images were captured under an inverted microscope. In group settings, according to the determination of adhesion, a serum-free medium containing the drug was added to the corresponding wells and incubated for 48 h for image capture under an inverted microscope. The quantification of the scratched area was performed at different time points using ImageJ1.0 software. The quantification of the scratched area was completed using formulas to calculate cell migration.
$$Healing\;rate\left(\%\right)=\frac{0\;h\;scratch\;distance-48\;h\;Scratch\;distance}{0\;h\;Scratch\;distance}\;\times \;100\%$$

#### 4.3.6. Statistical Analysis

Experimental data were analysed and plotted using GraphPad Prism 9.0 software. Statistical analysis was performed using one-way ANOVA and Tukey’s multiple comparison test. The experimental data were expressed as mean ± standard deviation (x ± s). Differences were considered statistically significant at *p* < 0.05.

## 5. Conclusions

In summary, we designed and synthesised two series of 16 hyperoside derivatives and assessed their cytotoxicity. Subsequently, we induced A549 cells with TGF-β1 to construct an in vitro lung fibrosis model, utilizing changes in cell phenotype during the epithelial–mesenchymal transition (EMT) as a criterion. Through qRT-PCR, microscopy, marker gene determination, and Western blot analyses, we identified eight derivatives exhibiting superior efficacy to Vimentin, with potential anti-EMT activity, all of which could significantly increase the expression of E-cadherin and decrease the expression of Vimentin in A549 cells. In addition, compound Z6 is currently considered to be the most potent EMT-inhibiting derivative of hyperoside, and the potentially active compounds were later validated using our cellular function assays and other means. The results showed that Z6 could enhance the cell adhesion ability, reduce the cell migration ability, and significantly inhibit the development of the EMT process. We utilised hyperoside as the lead compound, retained its basic backbone structure, and modified and optimised it. Z6 has improved solubility and bioavailability when compared to the prototype compound. This study initially investigated its derivatives and their possible pharmacological properties. It also provides relevant experimental data and a theoretical basis for the research and development of flavonoid derivatives as target drugs for anti-EMT therapy.

## Data Availability

Data included in the article/Appendix A.

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
