# Peer review of "Structural Modification and Optimisation of Hyperoside Oriented to Inhibit TGF-β-Induced EMT Activity in Alveolar Epithelial Cells"

_pharmaceuticals, 2024, doi:10.3390/ph17050584_

Round 1
Reviewer 1 Report
Comments and Suggestions for Authors
The manuscript is quite well written. The topic is interesting. I have some comments:
1) Keywords: Hypericin derivatives; Pulmonary fibrosis; TGF-β1; Epithelial-mesenchymal transi- 39 tion(EMT). Please add some keywords. This can provide readers with a quick overview of the research.
2) Human lung cancer A549 cells are of type II alveolar epithelial cell origin. They can 91 be used as a commonly used in vitro model to study the process of lung fibrosis after 92 inducing epithelial-mesenchymal transition using TGF-β1 [25-27]. Therefore, we have 93 modified the structure of hyperoside as a lead compound, synthesized a series of deriva- 94 tives, and evaluated their cytotoxicity effects in A549 cells in vitro. In addition, we identi- 95 fied the potential of hyperoside derivative (Z6) in reversing TGF-β1-stimulated epithelial... Please underline the study aim and add some information regarding the novelty of the study.
3) 4.3.6 Statistical analysis 533 Experimental data were analyzed and plotted using GraphPad Prism 9.0 software. 534 The experimental data were expressed as mean ± standard deviation (x±s). One-way 535 analysis of variance (ANOVA) was used to compare between groups, and the two-way 536 comparison was performed by the LSD test. Differences were considered statistically 537 significant at P<0.05. Please, improve this paragraph to clarify the tests used to evaluate the data.
4) 5. Conclusions 539 In summary, we designed and synthesized two series of 16 vimentin derivatives 540 and determined the cytotoxicity of the 16 synthesized compounds. Then we constructed 541 an in vitro lung fibrosis model by inducing A549 cells with TGF-β1 and used the change 542 of cell phenotype in the process of EMT as an entry point, and then determined eight 543 derivatives with better efficacy than vimentin by qRT-PCR microscopy, marker gene 544 determination and Western blot, and identif ..
I suggest to underline the novelty of the study and the possible clinical implication.
Comments on the Quality of English LanguageMinor changes of English language are required
Reviewer 2 Report
Comments and Suggestions for Authors
I had the opportunity to review this study investigating the role of new compounds in regulating EMT in A549 cells after TGF-b stimulation. I have several suggestions.
Major:
1. The selection of hyperoxide derivatives should be presented in more detail. The refererence of Heliyon couldn't show the process why hyperoside derivative was chosen as the target drugs.
2. Pulmonary fibrosis is mainly through dysregulation of fibroblast. Authors should also investigate the role of these compounds on fibroblasts.
3. The expression of collagen should also be presented.
4. Dose response of these compounds should be presented.
5. EMT could be presented by iimmunofluorescence double staining of E-cadherin/vimentin to better depict the process of EMT.
Other:
1. Many typos in errors in grammar and presentations. Authors should check their manuscript more carefully.
2. Methods 4.3.2 described SB-431542. However, SB-431542 is not used in this study.
3. Individual data dots should be added on the bar chart.
Comments on the Quality of English LanguageEnglish editing is mandatory.
Reviewer 3 Report
Comments and Suggestions for Authors
In this study, flavonoid hypericin and a series of hyperoside derivatives were synthesized by modifying the structure of hyperoside, and these compounds were then evaluated for their effects on epithelial-mesenchymal transition (EMT) in A549 cells, an important process contributing to pulmonary fibrosis. The effects of these compounds on A549 cytotoxicity, EMT-associated cell adhesion, migration, and biomarker expression were assessed. The design of the study is reasonable, however, there are quite a few major issues for the authors to solve.
1. The biggest problem I found was the short of DMSO control. The authors stated 40 mM stock concentration of compounds were diluted to a working concentration of 20 uM, which means a DMSO concentration of 0.05%. A corresponding DMSO control must be used for all the experiments for comparison.
2. N=3 was too low, and not appropriate to run statistical analysis.
3. Experimental procedures were not detailed enough. For example, when were the drug and TGFbeta given, respectively? Was it pre-treatment or post-treatment? In the adhesion and migration protocols, no drug or TGF-beta was even mentioned.
4. What's the underlying mechanisms mediating anti-EMT effect? The authors should check relevant signaling molecules (Smad2/3 etc) or transcription factors (Snail/Slug, etc).
Comments on the Quality of English LanguageThe manuscript needs extensive English editing.
Round 2
Reviewer 2 Report
Comments and Suggestions for Authors
Although authors revised the manuscript. I still have concerns about following issues.
1. The reason for selection of hyperoxide derivatives as potential agent is not explained.
2. The effect of these compounds in fibroblast, collagen and other ECM were not explored. It's fundametal for a potential antifibrotic agent.
3. Dose reponse data were still not presented.
Reviewer 3 Report
Comments and Suggestions for Authors
I have no more comments.